# Brief Communication: Early season snowpack loss and implications for over-snow vehicle recreation travel planning

Benjamin J. Hatchett[1,2], Hilary G. Eisen[3]

[1]Division of Atmospheric Sciences, Desert Research Institute, Reno, Nevada, 89512, USA
[2]Western Regional Climate Centre, Desert Research Institute, Reno, Nevada, 89512, USA
[3]Winter Wildlands Alliance, Boise, Idaho, 83702, USA

*Correspondence to*: Benjamin J. Hatchett (benjamin.hatchett@gmail.com)

**Abstract.** Over-snow vehicle recreation contributes to rural economies but requires a minimum snow depth to mitigate negative impacts on the environment. Daily snow water equivalent (SWE) observations from weather stations in the Lake Tahoe region (western USA) and a SWE reanalysis product are used to estimate the onset dates of SWE corresponding to ~30 cm snow depth ($SWE_{min}$). Since 1985, median $SWE_{min}$ onset has shifted later by approximately two weeks. Potential proximal causes of delayed onset are investigated; rainfall is increasing during October-January with dry days becoming warmer and more frequent. Adaptation strategies to address over-snow vehicle management challenges in recreation travel planning are explored.

## 1 Introduction

Ongoing and projected climate change is accelerating the warming of the cryosphere throughout Earth's mountain regions (Huss et al., 2017). Reductions in winter season snow, ice, and permafrost cover and volume primarily result from rising air temperatures (Brown and Mote, 2009) and shifts in precipitation from snow to rain (McCabe et al., 2018). These changes have cascading effects from mountains to lowlands with wide-ranging socioeconomic and ecologic impacts (Huss et al., 2017). In mountain regions of the United States, Europe, and Canada, winter recreation and tourism are central to economic activity. The economic benefits from winter recreation are projected to decline as a result of continued climate change that reduces season length and makes access to reliable snow more difficult (McBoyle et al., 2007; Scott et al., 2007; Wobus et al., 2017; Steiger et al., 2017).

Most winter tourism-based climate change impact studies have focused on ski resort-related activity (Steiger et al., 2017), although research has begun to address how other recreation-based components of the winter economy may be affected (McBoyle et al., 2007; Scott et al., 2007; Tercek and Rodman, 2016; Wobus et al., 2017, Hagenstad et al. 2018). Skier visits are positively correlated to snowfall (Hagenstad et al., 2018) and we assume that such a correlation is consistent across winter recreation activities. Due to the dependence on natural snowfall and reduced adaptive capacity compared to the ski community, which can use cost-effective snowmaking to augment the natural snowpack, over-snow vehicle (OSV)

recreation is highly vulnerable to climate variability and change (McBoyle et al., 2007; Scott et al., 2007). Climate change projections for Canada and the northeastern United States under an aggressive greenhouse gas emissions scenario suggest that by the mid-21[st] century, OSV season lengths will be reduced by 50-100% in most areas (McBoyle et al., 2007; Scott et al., 2007). A survey of the OSV community in Vermont found that reductions in the length of the winter season with sufficient snow coverage for OSV use were observed by 45% of respondents, with 74% of respondents decreasing their OSV use in response to low snow conditions (Perry et al., 2018). This survey also found that encounters with other recreationalists, including OSV users, detracted from a high-quality recreation experience. The net effects of reduced season length, more congestion, and lower quality experiences result in lower economic benefits from consumer surplus, or the amount a person is willing to pay over the amount actually spent. For OSVs, consumer surplus is estimated to be approximately 61 USD person$^{-1}$ day$^{-1}$ (Hagenstad et al., 2018).

In the Lake Tahoe region of California (Figure 1a), and many other rural mountain areas of the western United States, OSV use is a regionally significant component of winter season recreation. Estimates of annual economic impact from OSV recreation in the United States range between 7 and 26 billion USD (Fassnacht et al., 2018). As a result, OSV recreation has an appreciable economic impact on rural counties within the northern Sierra Nevada, many of which have a greater dependence on tourism-related employment than elsewhere in California (United States Census, 2013).

The proximity of the Lake Tahoe region to large population centres creates demand for OSV recreation over a limited and ecologically sensitive area. In order to limit potential negative impacts on natural resources (e.g., Keddy et al., 1979) during OSV operation, a minimum snow depth must be present. Minimum snow depth restrictions have been proposed by several forests undergoing winter travel management planning across the Sierra Nevada. This restriction is usually proposed as a minimum depth of 30 cm of un-compacted snow (United States Forest Service (USFS), 2013). Few forests have such a requirement at this time, but several are currently engaging in the process of winter travel management planning in response to a 2015 U.S. Federal Court ruling (Federal Register, 2015). The Eldorado National Forest in northern California (located in the southwestern quadrant of the study area) currently requires a minimum snow depth of approximately 30 cm for off-trail OSV use.

To our knowledge, no precise value of this minimum depth has been established through comprehensive studies quantifying OSV use and impacts or disturbance. Nonetheless, evidence indicates that OSV use can alter the landscape when a shallow snowpack is present. Keddy et al. (1979) observed that OSV use on very shallow snow (10-20 cm deep) doubled snow density and compressed underlying vegetation. When OSV use began under a deeper snowpack, less difference in snow density and hardness was observed compared to a control (no-OSV use) snowpack (Fassnacht et al., 2018). Further complicating the minimum depth requirement is the dependence of snow depth on the density of the snow, which varies seasonally and as a function of weather conditions that drive snowpack metamorphism processes (Sturm et al., 2010).

Resource managers tasked with day-to-day operations such as opening and closing OSV trailheads over large, diverse areas may not have the resources to visit trailheads to obtain snow depth and density measurements. Instead, they often rely on subjectively-based qualitative assessments of what is deemed sufficient snow. Managers often do not set a specific OSV season, leaving it to user discretion to determine when OSV use is appropriate. This can potentially cause conflict with other uses during the start and end to the winter season and can allow opportunities for inadvertent damage to natural resources due to insufficient snow depth. Here, we estimate the median timing of achieving sufficient snow depths for OSV operation and their trends during the past 34 years using observations of snow water equivalent (SWE) and a reasonable assumption of snow density. We focus on the initial timing of sufficient snow depth since the greatest demands for OSV recreation and potential ecological impacts occur between early and middle winter. The proximal causes of the identified increasingly later onset of achieving a minimum SWE value are further investigated. Because the trend towards later onset is not expected to reverse under continued regional warming, we provide adaptation strategies to cope with diminishing early season snowpack resources that can be included in forest travel management plans. The techniques can be extended to other regions where OSV recreation is an important component of economic activity and where early winter snowpack losses may be impacting seasonal recreation.

## 2 Data and Methods

The study area is the Lake Tahoe region of the western United States, a coastal, moderate elevation snow-dominated mountain range (Figure 1a). Daily maximum and minimum temperature, SWE, and precipitation were acquired for 16 SNOTEL stations from the Natural Resource Conservation Service (http://www.nrcs.gov/snotel). Daily, gridded estimates of SWE at 100 m horizontal resolution were provided by a satellite-era SWE reanalysis product (Margulis et al., 2015, 2016). The SWE reanalysis utilizes a Bayesian data assimilation framework to condition *a priori* snow model estimates with Landsat fractional snow-covered area images (Margulis et al., 2015). It verifies the posterior estimates against *in situ* daily snow pillow and monthly snow course data, which were found to compare favourably to previous studies (Margulis et al., 2016). The limitations of the remote sensing approach include lower temporal frequency of Landsat passes (approximately every 16 days) and potential obscuration of the land surface by clouds and vegetation, which can reduce usable imagery. Challenges with the *in situ* verification data include representativeness, or the discrepancies resulting from point-based snow pillow versus transect-based snow course SWE measurements, undersampling of forested and sloped terrain, and the bias of sites towards the intermediate elevations of the Sierra Nevada (50% of the stations are between 1500 and 2500 m; Margulis et al., 2016). The period studied encompasses October 1 1984 to March 31 2018 (2016 for the SWE reanalysis), which corresponds to the winter seasons of 1985-2018.

No accepted value of a minimum snow depth exists for OSV operation. Anecdotal values used by managers vary between 15-45 cm depending on compaction (USFS, 2013), but these do not take into account variability in snow density. To provide

a conservative and reasonable estimate of sufficient snow depth for what is assumed to be required for non-intrusive OSV operation, we specified 90 mm SWE (hereafter $SWE_{min}$) as the required minimum SWE corresponding to a minimum uncompacted depth of 30 cm for approval of OSV use. This value was obtained by equation (1):

$$SWE\ [mm] = d\ [mm] * {\rho_s}/{\rho_w}, \tag{1}$$

where $d$ is depth, $\rho_s$ is the density of the snow and $\rho_w$ is the density of water. We assume that in a coastal snowpack with marginal compaction, $\rho_s$ is typically 0.3 g/cm$^3$ (Sturm et al., 2010). This value appears reasonable to approximate a depth of 30 cm for early season conditions and is consistent with values used by the USFS (2013). Our $SWE_{min}$ value is close to Patterson (2016) and Tercek and Rodman (2017), who both chose 100 mm SWE as a threshold value for winter recreation in the Rocky Mountain National Park and Yellowstone National Park, respectively. We report the median timing of when each SNOTEL station and reanalysis gridpoint achieves $SWE_{min}$ and the annual timing as the median of the 16 SNOTEL stations.

To explore possible processes controlling the onset date of $SWE_{min}$, snow fractions ($S_f$) between October 1 and January 31 were calculated using the empirical hyperbolic tangent function formula developed by Dai (2008) with Sierra Nevada ecoregion parameter values estimated by Rajagopal and Harpold (2016). In contrast to Rajagopal and Harpold (2016), who used maximum temperature to estimate snow fraction, we selected average temperature because it gave a closer approximation to the mean snow level (~1,750 m) based upon independent estimates from observations (Hatchett et al., 2017). Dry days were days when less than the minimum measureable amount of precipitation (2.54 mm) was measured at SNOTEL stations. Mean minimum temperatures on dry days were calculated over the 16 stations for each year, as minimum temperature influences both snowpack dynamics and ecological processes (Oyler et al., 2015).

For all data, linear fits were estimated using a Theil-Sen slope and we report Spearman rank correlations. Statistical significance was tested using a modified Mann-Kendall test that accounts for serial correlation (see Hatchett et al., 2017 and references therein).

## 3 Results and Discussion

### 3.1 Timing of $SWE_{min}$

Median timing of achieving $SWE_{min}$ ranged from early November to early January and was negatively correlated with elevation ($R^2$=0.41, p<0.01; Figures 1a and 1b). For the selected $SWE_{min}$, nine of the 16 stations have significant (p<0.1) trends towards later onset of $SWE_{min}$ (Figure 1b). 13 of the 16 stations demonstrated a significant (p<0.1) trend when a value of $SWE_{min}$ between 80 and 100 mm was chosen (Figure 1b). There was no relationship between trend in onset date and elevation, which suggests that regional weather variability is a first-order control on snowpack conditions. At the regional level, the median trend across all stations was 0.6 day year$^{-1}$ (p<0.001; Figure 2a). This equates to $SWE_{min}$ being achieved approximately 20 days later between the present day and the beginning of the record, although interannual variability still

exists (Figure 2a). Results from the SWE reanalysis product are broadly consistent with the station-based analysis, indicating timing of $SWE_{min}$ is largely a function of elevation (Figure 1a). The median trend of the domain (approximately 15 days over the study period or 0.48 day year$^{-1}$) is close to the SNOTEL-based trend with the largest trends occurring above 2000 m (Figure 1c). The median trend of the domain when only considering statistically significant gridpoints ($p<0.05$) is approximately 21 days over the study period or 0.67 day year$^{-1}$ (Figure 1d). The consistency of the results between the independent SNOTEL data and the SWE reanalysis product support the hypothesis that a delayed onset of $SWE_{min}$ is occurring in the Lake Tahoe region. During years with later onset of $SWE_{min}$ (such as 1991, 2012, or 2014; Figure 2a) most OSV users would likely opt out of recreating during much of the season due to potential mechanical damage to their vehicles. However, if sufficient snow existed above a certain elevation, inadvertent damage to the landscape could result when OSVs travel over shallow snowpacks in order to reach destinations with deeper snow. To ensure access to higher elevation areas for OSV use during poor lower elevation snowpack conditions, management plans could identify and implement corridors or rights-of-way that minimize landscape impacts while allowing access (Table 1).

### 3.2 Possible drivers of timing changes of $SWE_{min}$

The increasingly later onset of $SWE_{min}$ (Figures 1c, 1d and 2a) is consistent with an observed increase (0.22 days yr$^{-1}$, $p<0.0001$) in the number of dry days during early winter (October-January; Figure 2b). Minimum temperatures on dry days are also increasing (0.098 °C yr$^{-1}$, $p<0.0001$). The observed decreasing trend towards reduced early season snow fraction ($S_f$; 0.66% year$^{-1}$, $p<0.0001$; Figure 2c), implies increasing numbers of warmer dry days and a shift towards increased rainfall are likely contributing to later onset of $SWE_{min}$. The reduction in precipitation falling as snow is primarily driven by warming temperatures (McCabe et al., 2018), which may be controlled by regional atmospheric and oceanic circulations that favour higher snow level storms (Hatchett et al., 2017). The higher snow levels (and hence lower $S_f$; Figures 2a and 2c) reduce snowpack accumulation during precipitation events and can allow for snowpack loss due to turbulent heat fluxes and heat input by rain. The more frequent and warmer dry conditions create additional opportunities during which snowpack loss can occur via radiative and turbulent fluxes. The analysis of SNOTEL temperature is limited by inhomogeneities introduced by temperature-dependent sensor biases leading to over-estimation of trends (Oyler et al., 2015). While over-estimation is greatest at elevations above 3000 m, additional assessments are needed to validate the robustness of the role of regional warming in reducing early season snowpack.

### 3.3 Implications for regional winter travel management planning

Due to its moderate elevation, the Lake Tahoe region is susceptible to climate change-induced warming (Walton et al., 2017). Our results provide another metric (later onset date of $SWE_{min}$) that is consistent with observations of ongoing changes in the Sierra Nevada cryosphere, including rising winter snow levels (Hatchett et al., 2017) and snowpack declines (Mote et al., 2018). Climate model projections for California support the continuation of these trends, with a drying and warming of the fall season (Swain et al., 2018) and an increased frequency of dry days (Polade et al., 2015). Projected snow-

covered area declines are estimated to be the greatest during the beginning and end of the snow season (Walton et al., 2017). As a result, forest travel management plans should include adaptation strategies (Table 1) that can help managers and recreators cope with the increasing chances of a later opening date for OSV use but also provide flexibility in the event of an early, snowier-than-normal start to the winter. Flexible strategies developed by diverse stakeholder groups through public discourse are encouraged, as the continued reduction of area available for motorized and non-motorized users will lead to increasingly frequent use conflicts if not addressed. More frequent use conflicts, particularly at trailheads or in congested areas, may lead to decreases in high quality experiences (Perry et al., 2018) and contribute to declines in OSV or other forms of recreational usage that reduce positive economic impacts (Hagenstad et al., 2018).

Developing a suite of adaptive management strategies is essential if land managers are to meet legal obligations to manage OSV recreation in a manner that minimizes impacts to natural resources, wildlife, and conflict between uses (Federal Register, 2015). As snow seasons become more variable and less dependable overall, it will be necessary to utilize several complementary management strategies if land managers want to continue to provide high quality opportunities for all forms of winter recreation. For example, setting season dates that encompass the general times of the year when OSV use is appropriate, paired with a minimum SWE (or snow depth, depending on data availability), and allowing for OSV use on certain routes with a lower snowpack to provide access to higher-elevation areas may help to extend the OSV season. Likewise, it may be necessary to relocate winter trailheads to higher elevations as areas with consistent snowpack become shifted upwards in elevation. As the strategies in Table 1 show, however, there are tradeoffs with any strategy and OSV recreation is not the sole use of public lands in winter. Managing OSV recreation must occur in concert with managing other forms of winter recreation and protecting wildlife and natural resources (Federal Register, 2015). There is no one-size-fits-all strategy that will work for every national forest. It is essential that land managers work with public and agency stakeholders to craft locally-appropriate and equitable adaptation measures, taking into account potential impacts to and conflicts with other recreation uses, wildlife, natural resources, and other land management goals. It may also be necessary to accept that in the future, OSV and other forms of winter recreation (e.g., backcountry skiing and snowshoeing) will not be supported across all of the areas where it historically occurred. Winter travel planning is thus an excellent opportunity for land managers, particularly the United States Forest Service, to proactively address OSV management and consider how climate change is affecting OSV activities on national forests in order to maintain the opportunity for this form of winter recreation and its positive economic impact.

## 4 Concluding Remarks

Using snow water equivalent and a density assumption as a proxy for depth, we have presented a pilot study aimed at a better understanding of when the Lake Tahoe region attains sufficient snowpack depth to allow safe over-snow vehicle (OSV) usage. A station-based analysis of 16 remote weather stations in the region and a spatially distributed SWE reanalysis product indicated that the median timing of achieving sufficient depth varies with elevation from early November to late

December. The median timing of sufficient depth has increased by approximately two weeks during the past three decades with significant changes on the order of three weeks. The proximal causes for this shift towards later onset appear to be due to both a shift from snowfall to rainfall and increases in dry day frequency and temperature during the early winter season. However, further research is needed to estimate specific contributions from each cause and constrain the role of surface-albedo and/or humidity feedbacks at various elevations throughout the region (Patterson, 2016; Walton et al., 2017).

A primary limitation of our study is the lack of an established snow depth to avoid negative impacts of OSV operation as a function of land cover type and snow density. The work of Fassnacht et al. (2018) represents an important advance towards achieving this value, which can be used to guide winter travel management planning, although the United States Forest Service has begun to recommend a depth (USFS, 2013). Additional studies on achieving regionally-relevant minimum snow depths and better quantification of economic and ecological impacts from reduced snow cover area and duration will guide more robust travel management plans in national forests. They also can help prioritize pragmatic adaptation strategies for specific regions. Given the economic impact of OSV recreation and likely reduction in land available for OSV or other human-powered recreation uses (McBoyle et al., 2007; Scott et al., 2007; Tercek and Rodman, 2016; Hagenstad et al., 2018) combined with increasing numbers of winter recreation participants (Fassnacht et al., 2018), achieving winter travel management plans that are adaptive to varying snowpack conditions while minimizing user conflicts will be a key step towards sustainable mountain recreation.

**Code Availability**

The MATLAB code used for analysis is available upon request.

**Data Availability**

All data is publically available and has been properly cited in the text.

**Competing Interests**

HGE is employed by the Winter Wildlands Alliance (WWA). BJH has consulted for the WWA.

**Author Contributions**

BJH and HGE conceived and designed the study, interpreted the results, and wrote the paper. BJH acquired data and performed the analysis.

## Acknowledgements

The project described in this publication was supported by Grant Number G14AP0076 from the United States Geological Survey (USGS). Its contents are solely the responsibility of the authors and do not necessarily represent the official views of the USGS. This manuscript is submitted for publication with the understanding that the United States Government is authorized to reproduce and distribute reprints for governmental purposes. We greatly appreciate the constructive review comments by Glenn Patterson, Daniel Scott, and Editor Ross Brown that helped us improve the quality of this manuscript.

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

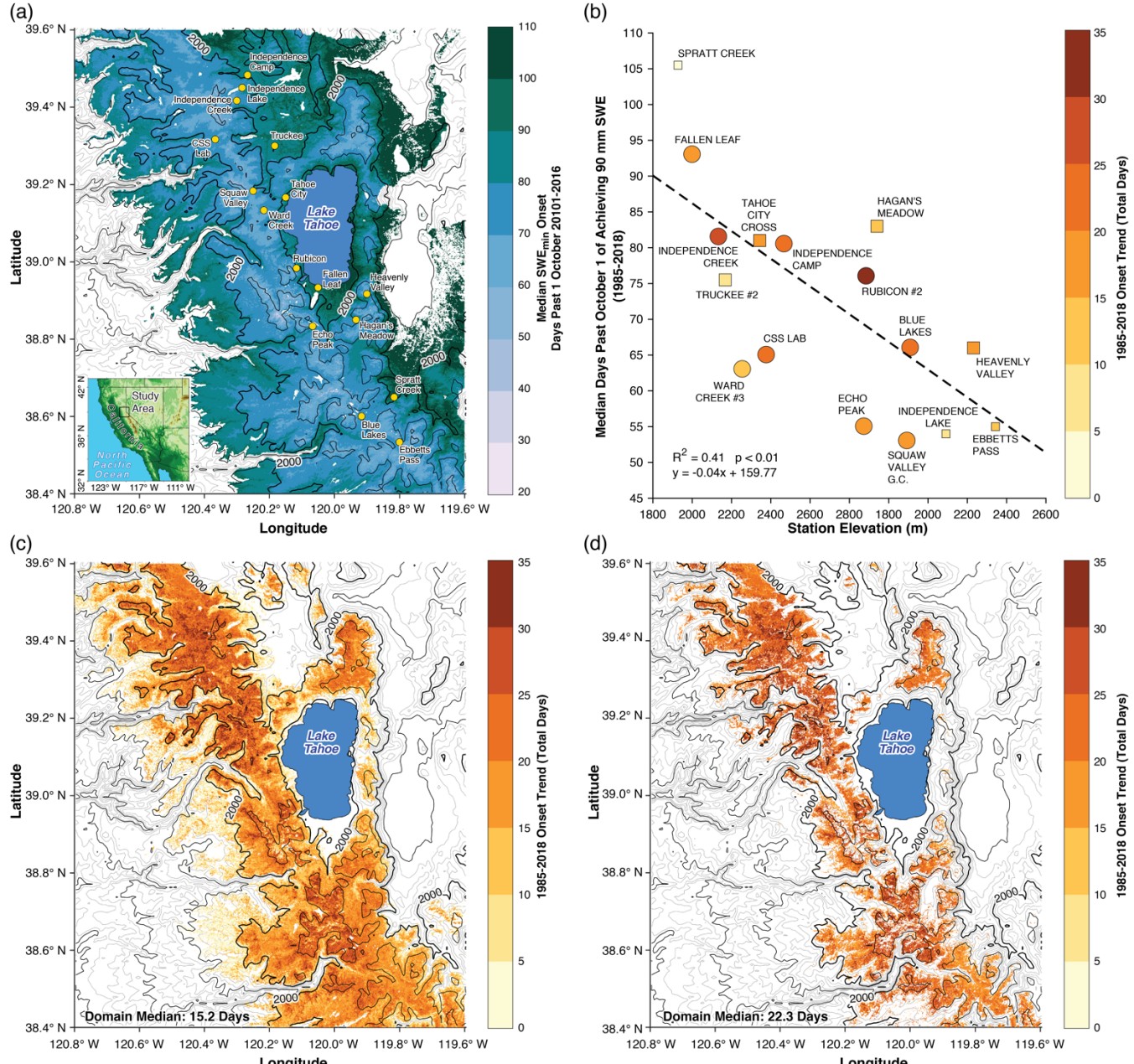

**Figure 1: (a)** Median 2001-2016 SWE$_{min}$ (days past October 1) based on the SWE reanalysis product (Margulis et al., 2016) with SNOTEL stations shown as gold dots. The inset map shows the study area. **(b)** Timing of median SWE$_{min}$ (days past October 1) by SNOTEL station elevation. Dots are colored by the trend (annual rate of snow depth timing change times 34 years). Dashed black line denotes the Theil-Sen linear fit. Large circles indicate significant trends (p<0.1) for SWE$_{min}$, while large squares indicate a significant (p<0.1) trend in SWE$_{min}$ was identified for a value of SWE$_{min}$ between 80 and 100 mm. Small squares indicate no significant trend. **(c)** Spatially distributed Theil-Sen linear trends in SWE$_{min}$ over the period 1985-2016, calculated as the annual rate times the 32-year period. **(d)** As in (c) but showing only gridpoints with a statistically significant (p<0.05) trend in onset date. In panels a, c, and d, the thin (thick) grey contour lines indicate elevations every 125 m (500 m) while the thick black line indicates the 2000 m elevation contour (labeled). Gridpoints with more than three missing years were excluded from the analysis.

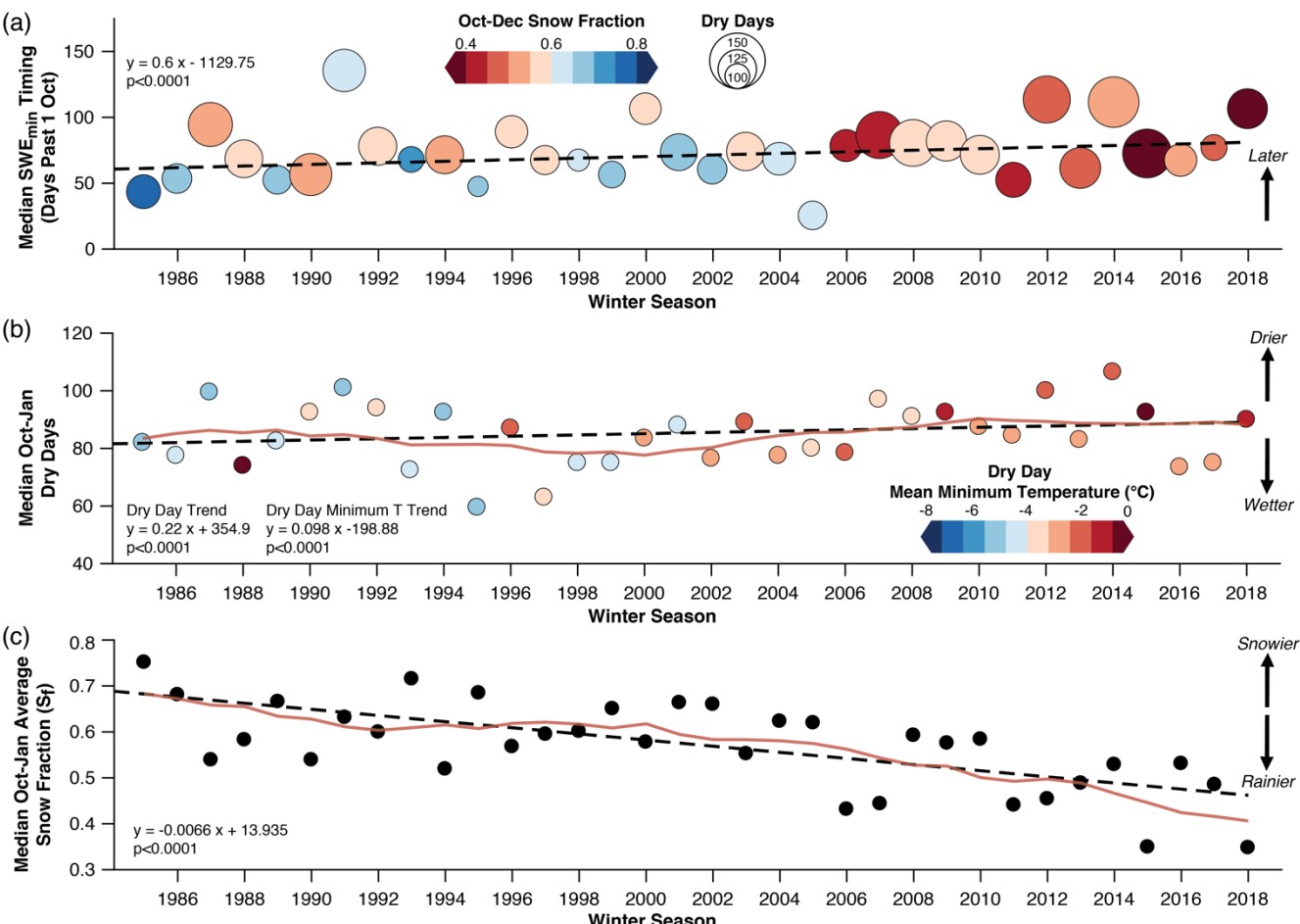

**Figure 2: (a) Annual median timing of SWE$_{min}$ (days past October 1) with dots colored by median October-January average snow fraction and sized according to the median number of October-January dry days. (b) Median early season (1 October-31 January) dry days with dots colored by average October-January minimum temperature. (c) As in (b) but for median snow fraction averaged over the 16 stations. In all figures, the dashed lines demonstrate Theil-Sen linear fits and red lines (b and c) show the five-year running mean.**

| Adaptation Measure | Benefit(s) | Challenge(s) |
|---|---|---|
| *Requirement of minimum snow depth off trail, but not on roads/marked trails, or a lower minimum snow depth on roads/marked trails* | Allow OSV use even under extremely low snow conditions, limits resource damage in wildlands; grooming could be utilized to maximize snow depth on road | Preventing users from going off trail under low snow conditions; enforcement, resources required to obtain snow condition information |
| *Ensure high elevation access via a right-of-way* | During warmer/drier years, snow conditions are likely to be better (deeper snowpack) at higher elevation | User group conflicts; presence of Wilderness at high elevation; impacts to snow-dependent wildlife species; demand; parking |
| *Removal of blanket opening dates* | Prevents opening before $SWE_{min}$ achieved and will limit damage to landscape | Resources required to obtain snow condition information |
| *Identify corridors that collect/retain more snow* | During otherwise poor snow conditions, these areas may allow OSV recreation to occur, particularly at lower elevation areas | Need for data on these corridors |
| *Improve durability of trailhead and corridor trails* | Allows OSV recreation to occur when minimal snow exists thereby reducing negative impacts in high-use areas | Need for specific quantification of how to improve durability; potential permitting problems |
| *Trade-off: closure of low elevation/sensitive habitat for improved high elevation access* | Eliminate chance of damaging landscapes in low elevation regions, increase in the number of days/year that OSV recreation can occur by enhanced high elevation access | Need for collaboration between stakeholders/user groups to identify areas where compromise could occur; may be opposed by those who must travel much further for OSV use |
| *Fee increases to enhance access and offset impacts from higher demand (i.e., restoration projects)* | Would provide for additional resources to monitor trailhead conditions, improve parking/bathrooms at trailheads, fund restoration projects and creation of low-snow OSV trails | Fees are generally opposed by members of the public |
| *Additional grooming* | Allows additional area for OSV use when conditions are insufficient for off-trail use | Costs for grooming equipment and personnel, many OSV users are primarily interested in off-trail use |
| *Clear designation of non-motorized areas (i.e., signage)* | Reduces user conflicts by improving knowledge and awareness of areas open (or closed) to OSV use | Costs related to enforcement as well as installation and upkeep of signage |

**Table 2: Adaptation strategies to address loss of early winter snowpack for OSV recreation.**