# Peer review of "Brief Communication: Early season snowpack loss and implications for over-snow vehicle recreation travel planning"

_The Cryosphere, 2018_

## Referee Comment (RC1) · G. Patterson (Referee) · 4 Sep 2018

Thank you for the opportunity to review this brief communication. In general I found the paper to be well-written, representing a significant contribution to our understanding of climate change, snowpack trends, and winter recreation. My only substantive suggestions are: (1) Based on figure 1(b) and on intuition, median timing of achieving SWEmin appears to be negatively, not positively, correlated with elevation. (2) I realize that it might not be feasible to add another parameter to the analysis at this late date, and I also realize that SNOTEL temperature records can be suspect, but I still feel it's worth bringing up this point. As you state well on page 4, line 23, warming temper-

atures are likely to be an important driver for the trends toward later SWEmin, and are also likely to be the primary factor involved in the shift toward more rain and less snow. Would it be possible to add a figure showing the trend in a relevant measure of temperature? It could be something like average daily (or max daily) early cold-season (Oct-Dec) temperature, or something like that. From what you are describing, one would expect to see a shift from predominantly freezing to melting temperatures. (3) I like the discussion pertaining to table 1, and the suggestions contained in the table. I would like to suggest one additional management strategy. As both human-powered and OSV winter recreational use are likely to be confined in the future to smaller geographic areas and shorter-duration seasons, conflicts among user groups are likely to increase. In the Rabbit Ears Pass area of Colorado, managers from Routt National Forest minimize those conflicts by clearly designating separate areas for motorized and non-motorized use (non-motorized can actually occur in both areas). In this case it is the east side and the west side of the pass. My suggestion is to develop such conflict-reduction strategies for the Lake Tahoe area, as well.

Below are some specific suggestions for minor editorial issues. Abstract: In the first sentence (line 8), it would be good to clarify whether "negative impacts" refer to impacts on the over-snow vehicle recreational experience, or on the environment. Based on the paper, it appears to be the latter. I agree with the previous suggestion, that it would be good to speak (line 11) in terms of later onset of SWE sufficient for recreation, rather than increasing date. Introduction: Page 1 Line 15: I would be a little more comfortable with "warming" of the cryosphere, rather than "decline". "Warming" is clear; "decline" carries complex connotations. Line 27: "Economic revenue" strikes me as redundant. I would suggest either "revenue" or "economic impact". Also, it would be helpful to specify that the revenue is on an annual basis: "...annual revenue..." Page 2 Lines 4-9: This would be a good place to address one of the important aspects of the SWE-depth relationship that is central to this discussion. I would suggest stating that the 30-cm minimum depth refers to 30 cm of uncompacted or fresh snow, something like this: "Minimum snow depth restrictions have been proposed by several

forests undergoing winter travel management planning across the Sierra Nevada. This restriction is usually proposed as a minimum depth of 30 cm of uncompacted snow (United 5 States Forest Service (USFS), 2013)" Data and Methods: Page 3 Line 9: This is a little nit-picky, but this sentence seems to indicate that SWE is a measure of depth. I would suggest rewording to: "…we specified 90 mm SWE (hereinafter SWEmin) as the required minimum SWE corresponding to a minimum uncompressed depth of 30 cm for approval of OSV use." Line 14: I agree with the earlier suggestion to maintain consistency of depth units by sticking with cm instead of mm. Line 24: I would suggest using "when" instead of "that": "Dry days were days when zero precipitation was measured…" Results and Discussion: Page 3 Lines 31-31: Based on figure 1(b) and on intuition, median timing of achieving SWEmin appears to be negatively, not positively, correlated with elevation. Page 4 Line 1: Delete unneeded "in" after "trends". Line 8: Thanks for looking at the elevation dependency of the SWEmin trends. I, too, found indications of elevation-dependent warming, meaning steeper trends at higher elevations. Some of the references in my 2016 dissertation go into this in a little more detail. Line 23: I realize that it might not be feasible to add another parameter to the analysis at this late date, and I also realize that SNOTEL temperature records can be suspect, but I still feel it's worth bringing up this point. As you state well here, warming temperatures are likely to be an important driver for the trends toward later SWEmin, and are also likely to be the primary factor involved in the shift toward more rain and less snow. Would it be possible to add a figure showing the trend in a relevant measure of temperature? It could be something like average daily (or max daily) early cold-season (Oct-Dec) temperature, or something like that. From what you are describing, one would expect to see a shift from predominantly freezing to melting temperatures. Table 1–Page 11: I like the discussion pertaining to table 1, and the suggestions contained in the table. I would like to suggest one additional management strategy. As both human-powered and OSV winter recreational use are likely to be confined in the future to smaller geographic areas and shorter-duration seasons, conflicts among user groups are likely to increase. In the Rabbit Ears Pass area of Colorado, managers

from Routt National Forest minimize those conflicts by clearly designating separate areas for motorized and non-motorized use (non-motorized can actually occur in both areas). In this case it is the east side and the west side of the pass. My suggestion is to develop such conflict-reduction strategies for the Lake Tahoe area, as well.

Please also note the supplement to this comment:
https://www.the-cryosphere-discuss.net/tc-2018-181/tc-2018-181-RC1-supplement.pdf

---

## Editor Comment (EC1) · R. D. Brown (Editor) · 6 Sep 2018

Nice paper and a very useful contribution! A few comments for your consideration during the review process:

1. Please make sure to use units of cm for depth throughout the paper to avoid confusion with SWE in mm e.g. p. 3 line 14 depth is given as 300 mm instead of 30 cm.

2. I suggest you provide a brief description (method, resolution, caveats etc) for the Margulis et al. (2016) SWE reanalysis given this is one of your major data sources.

[Figure]

3. Your paper focusses only on SWEmin timing but the duration of SWE >= SWEmin would also be very relevant, and may be more critical in terms of economic impacts. I'm curious as to why you didn't include this as part of the study.

4. In your abstract I suggest you talk about delayed onset date of SWEmin rather than the date increasing... the later is not intuitive to many people.

Best regards, Ross Brown (ed)
* * *

---

## Referee Comment (RC2) · D. Scott (Referee) · 3 Nov 2018

I appreciate the opportunity to review this brief communication, as there is very limited work on the implications of changing cryosphere for the multi-billion dollar over-snow vehicle (OSV) ('snowmobile') industry. This industry is far more at risk to cryosphere changes than the ski industry, which has much higher adaptive capacity through cost-effective snowmaking. I concur with G. Patterson that the paper is very well written and the comments on methods and interpretation, and will build on those remarks.

The dual data sources are very important to enable the exportability of the method to other regional markets across North America (mostly via reanalysis, as Snotel stations

are limit in Eastern markets) that would allow inter-regional market comparisons.

A limitation to the paper is that literature review is not comprehensive and given how limited this literature is, it should be complete in my opinion. While I acknowledge space is limited in a brief communication, this will provide future authors with a complete and current state of knowledge to build on. This will also strengthen some of the threshold assumptions made in this paper and comparisons with other regional markets. Specifically, the authors should consider the work on snowmobile tourism under climate change that was completed in New England (Scott et al. 2008. Mitigation and Adaptation Strategies to Global Change, 13, 5-6, 577-59) and parts of Canada (McBoyle et al. 2007. Managing Leisure, 12, 4, 237-250) about 10 years ago.

Discussion of impacts for visitor experience or economic impacts could be strengthened. Analyses of the impacts of recent record warm winters on the ski industry have revealed that shorter, more varied seasons result in increased congestion, which has adverse impacts on visitor experience (and thus economic surplus). The same impact is likely with OSV (particularly at trailheads) if demand remains stable. Have recent record warm winters revealed any impacts on visitor use patterns or increased impacts on landscapes/ecology?

The adaptation table is very useful for resource managers to consider appropriate responses. I fully agree with the authors that there is no one-size-fits-all approach, and that climate adaptation has to be informed by local circumstances and stakeholders. Other options the might be included in this table could include: (1) improve smoothness/durability of trailhead and corridor trails, so to require less snow and reduce impacts; (2) restrict access to marked trail areas in early/late season or during mid-season low snow periods; (3) if trail groomers are not used in this region, introduce them to improve the durability of corridor trails.

---

## Author Comment (AC1) · 29 Nov 2018

**Author responses to: "Brief Communication: Early season snowpack loss and implications for over-snow vehicle recreation travel planning"**

Responses to reviewers are in **bold,** *new text is in italics* **(bold italics for emphasis)**

**Responses to Reviewer 1**

Review comments by Glenn Patterson on Hatchett and Eisen, 2018, Brief Communication: Early season snowpack loss and implications for over-snow vehicle recreation travel planning. Thank you for the opportunity to review this brief communication. In general I found the paper to be well-written, representing a significant contribution to our understanding of climate change, snowpack trends, and winter recreation.

**Dear Dr. Patterson,**
**We appreciate your constructive comments and positive feedback. Please find below responses and our revisions to address all of your comments.**

My only substantive suggestions are: (1) Based on figure 1(b) and on intuition, median timing of achieving SWEmin appears to be negatively, not positively, correlated with elevation.

**Thank you for pointing out this oversight. You are correct, the timing of SWEmin is negatively correlated with elevation. We have changed the text to "*negatively correlated*".**

(2) I realize that it might not be feasible to add another parameter to the analysis at this late date, and I also realize that SNOTEL temperature records can be suspect, but I still feel it's worth bringing up this point. As you state well on page 4, line 23, warming temperatures are likely to be an important driver for the trends toward later SWEmin, and are also likely to be the primary factor involved in the shift toward more rain and less snow. Would it be possible to add a figure showing the trend in a relevant measure of temperature? It could be something like average daily (or max daily) early cold-season (Oct-Dec) temperature, or something like that. From what you are describing, one would expect to see a shift from predominantly freezing to melting temperatures.

**We appreciate this useful suggestion. The snow fraction calculation is based upon average temperature and thus captures the shift from snow to rain you mention, but for non-precipitating (dry) days, we have added the average early season dry day minimum temperature value to Panel b on Figure 2. Minimum temperature is an excellent metric to observe how regional warming (via changes in longwave radiation balance driven by changes in atmospheric composition) is impacting the study area. The Mann-Kendall trend test was applied to the minimum temperature timeseries. It yielded a statistically significant trend of 0.098°C yr$^{-1}$ (p<0.0001). This suggests regional warming is reducing precipitation falling as snow on wet days and is driving a warming of an ever-increasing number of dry days, both of which will act to negatively force the snowpack. For consistency, we have calculated dry days,**

**average dry day minimum temperature, and snow fraction over the early season (mid fall through early winter) period spanning October-January. The main text and figure 2 have been updated with the adjusted results.**

**The new text is as follows:**

**The new figure 2 and caption:**

[Figure]

**We also agree that the SNOTEL temperature records can be suspect (based on the Oyler et al. 2015 work and have noted this in the text at the end of section 3.2 (possible drivers of SWE loss):**

*"The analysis of SNOTEL temperature is limited by inhomogeneities introduced by temperature-dependent sensor biases leading to over-estimation of trends (Oyler et al., 2015). While over-estimation is greatest at elevations above 3000 m, additional assessments are needed to validate the robustness of the role of regional warming in reducing early season snowpack"*

**Added reference:**
Oyler, J.W., Dobrowski, S.Z., Ballantyne, A.P., Klene, A.E. and Running, S. W.: Artificial amplification of warming trends across the mountains of the western United States, Geophys. Res. Lett., 42(1), 153-161, https://doi.org/10.1002/2014GL062803, 2015.

(3) I like the discussion pertaining to table 1, and the suggestions contained in the table. I would like to suggest one additional management strategy. As both human-powered and OSV winter recreational use are likely to be confined in the future to smaller geographic areas and shorter-duration seasons, conflicts among user groups are likely to increase. In the Rabbit Ears Pass area of Colorado, managers from Routt National Forest minimize those conflicts by clearly designating separate areas for motorized and non-motorized use (non-motorized can actually occur in both areas). In this case it is the east side and the west side of the pass. My suggestion is to develop such conflict-reduction strategies for the Lake Tahoe area, as well.

**Thank you for the suggestion, we have added this strategy to the table. From personal experience, although some areas are well-signed (designated), problems can still arise due to lack of resources to enforce closures, but such an issue should be addressed during the travel management planning process and through subsequent budget requests.**

Below are some specific suggestions for minor editorial issues.
Abstract: In the first sentence (line 8), it would be good to clarify whether "negative impacts" refer to impacts on the over-snow vehicle recreational experience, or on the environment. Based on the paper, it appears to be the latter.

**You are correct, it is the latter. We have added "*on the environment*" to clarify.**

I agree with the previous suggestion, that it would be good to speak (line 11) in terms of later onset of SWE sufficient for recreation, rather than increasing date.

**We have changed the text to reflect '*later onset*' rather than increasing date.**

Introduction:
Line 15: I would be a little more comfortable with "warming" of the cryosphere, rather than "decline". "Warming" is clear; "decline" carries complex connotations.

**We have changed the text to "*warming*".**

Line 27: "Economic revenue" strikes me as redundant. I would suggest either "revenue" or "economic impact". Also, it would be helpful to specify that the revenue is on an annual basis: "…annual revenue…"

**Thank you for the suggestions. We have changed the text to "*estimates of annual economic impact*".**

Lines 4-9: This would be a good place to address one of the important aspects of the SWE-depth relationship that is central to this discussion. I would suggest stating that the 30-cm minimum depth refers to 30 cm of uncompacted or fresh snow, something like this: "Minimum snow depth restrictions have been proposed by several forests undergoing winter travel management planning across the Sierra Nevada. This restriction is usually proposed as a minimum depth of 30 cm of uncompacted snow (United 5 States Forest Service (USFS), 2013)"

**Good point, we have changed the text to follow your suggested phrasing:**
*"…across the Sierra Nevada. This restriction is usually proposed as a minimum depth of 30 cm of un-compacted snow (United States Forest Service (USFS), 2013)."*

Data and Methods:

Line 9: This is a little nit-picky, but this sentence seems to indicate that SWE is a measure of depth. I would suggest rewording to: "…we specified 90 mm SWE (hereinafter SWEmin) as the required minimum SWE corresponding to a minimum uncompressed depth of 30 cm for approval of OSV use."

**Change has been made, thank you for helping to clarify this and hopefully reduce confusion. The new text is as follows:**
*"…we specified 90 mm SWE (hereafter $SWE_{min}$) as the required minimum SWE corresponding to a minimum un-compacted depth of 30 cm for approval of OSV use."*

Line 14: I agree with the earlier suggestion to maintain consistency of depth units by sticking with cm instead of mm.

**We are now using cm as the primary unit with respect to depth and mm for snow water equivalent. The text has been changed to "*30 cm*".**

Line 24: I would suggest using "when" instead of "that": "Dry days were days when zero precipitation was measured…"

**Change has been made to "*when*", thank you.**

Results and Discussion:
Lines 31-31: Based on figure 1(b) and on intuition, median timing of achieving SWEmin appears to be negatively, not positively, correlated with elevation.

**Change to "*negatively correlated*" has been made, again we apologize for the oversight.**

Line 1: Delete unneeded "in" after "trends".

**We have removed "in".**

Line 8: Thanks for looking at the elevation dependency of the SWEmin trends. I, too, found indications of elevation-dependent warming, meaning steeper trends at higher elevations. Some of the references in my 2016 dissertation go into this in a little more detail.

**While our station-based SWE analysis did not find a relationship between trend in onset date, the gridded SWE analysis did indicate more significant trends at higher elevations, however this is still likely due to the early season weather variability (as previously noted in the original submission) and the bias of reanalysis validation data (snow pillows and snow courses) being located at intermediate elevations.**

**We added some additional text to the concluding remarks suggesting further assessment of the controls of elevation-dependent warming (notably, changes in humidity):**
**"*However, further research is needed to estimate specific contributions from each cause and constrain the role of surface-albedo and/or humidity feedbacks at various elevations throughout the region (Patterson, 2016; Walton et al., 2017).*"**

Line 23: I realize that it might not be feasible to add another parameter to the analysis at this late date, and I also realize that SNOTEL temperature records can be suspect, but I still feel it's worth bringing up this point. As you state well here, warming temperatures are likely to be an important driver for the trends toward later SWEmin, and are also likely to be the primary factor involved in the shift toward more rain and less snow. Would it be possible to add a figure showing the trend in a relevant measure of temperature? It could be something like average daily (or max daily) early cold-season (Oct-Dec) temperature, or something like that. From what you are describing, one would expect to see a shift from predominantly freezing to melting temperatures.

**This is a great suggestion and we have added this analysis to the manuscript, despite the limitations of the SNOTEL temperature data. Please see the response to main suggestion 2) above.**

Table 1--Page 11:
I like the discussion pertaining to table 1, and the suggestions contained in the table. I would like to suggest one additional management strategy. As both human-powered and OSV winter recreational use are likely to be confined in the future to smaller geographic areas and shorter-duration seasons, conflicts among user groups are likely to increase. In the Rabbit Ears Pass area of Colorado, managers from Routt National Forest minimize those conflicts by clearly designating separate areas for motorized and non-motorized use (non-motorized can actually occur in both areas). In this case it is the east side and the west side of the pass. My suggestion is to develop such conflict-reduction strategies for the Lake Tahoe area, as well.

**Thank you for the suggestion, we have added this strategy to Table 1.**

---

## Author Comment (AC2) · 29 Nov 2018

**Author responses to: "Brief Communication: Early season snowpack loss and implications for over-snow vehicle recreation travel planning"**

Responses to reviewers are in **bold,** *new text is in italics* **(bold italics for emphasis)**

**Responses to Editor Brown**

Nice paper and a very useful contribution! A few comments for your consideration during the review process:

**Dear Dr. Brown,**
**We appreciate your positive feedback and suggested revisions. Please find below responses and our revisions to address all of your comments.**

1. Please make sure to use units of cm for depth throughout the paper to avoid confusion with SWE in mm e.g. p. 3 line 14 depth is given as 300 mm instead of 30 cm.

**Thank you for the suggestion. We have edited the text to "*30 cm*"**

2. I suggest you provide a brief description (method, resolution, caveats etc) for the Margulis et al. (2016) SWE reanalysis given this is one of your major data sources.

**Good point, we have added the following discussion regarding the SWE reanalysis:**
*"The SWE reanalysis utilizes a Bayesian data assimilation framework to condition a priori snow model estimates on Landsat fractional snow-covered area images (Margulis et al., 2015). It verifies the posterior estimates against in situ daily snow pillow and monthly snow course data and is shown to compare favourably to previous studies (Margulis et al., 2016). The limitations of the remote sensing approach include lower temporal frequency of Landsat passes (approximately every 16 days) and potential obscuration of the land surface by clouds and vegetation. Challenges with the in situ verification data include representativeness, or the discrepancies resulting from point-based snow pillow versus transect-based snow course SWE measurements, undersampling of forested and sloped terrain, and the bias of sites in the intermediate elevations of the Sierra Nevada (50% of the stations are between 1500 and 2500 m; Margulis et al., 2016)."*

**The initial submission did include a sentence on the resolution in time and space of the reanalysis (see below in bold); this sentence precedes the newly added discussion on the method above. We did add the additional citation of the previous Margulis work for readers who might be interested in this approach.**

"**Daily**, gridded estimates of SWE at **100 m horizontal resolution** were provided by a satellite-era SWE reanalysis product (Margulis et al., **2015**, 2016)."

**Added reference:**
Margulis, S., Girotto, M., Cortés, G., and Durand, M.: A particle batch smoother approach to snow water equivalent estimation, J. Hydrometeor., 16, 1752–1772, doi:https://doi.org/10.1175/JHM-D-14-0177.1, 2015.

3. Your paper focusses only on SWEmin timing but the duration of SWE >= SWEmin would also be very relevant, and may be more critical in terms of economic impacts. I'm curious as to why you didn't include this as part of the study.

**Thank you for bringing this up. We did consider this, but wanted to focus on the early season onset as this is when (anecdotally) the desire to recreate on snow is the highest and when the snow depths are most likely to be at the threshold of insufficient coverage for safe OSV usage to avoid damage to the landscape. In other words, the greatest demand for OSV recreation coincides with the early season (holiday periods, excitement to recreate on snow) and in all but the worst snow years, the ability to recreate is not limited by lack of snow in the spring season. In most years (and especially poor snow years), interest wanes during the late season before lack of snow limits OSV recreation. The best economic impacts will likely be during the early-middle season as people are excited to purchase new equipment and utilize holiday periods for extended vacations. As the snow-covered area retreats uphill during the spring season, most users opt to enjoy higher elevation regions anyways given the favorable weather conditions and greater snow stability. The end of season SWE timing can be related to numerous and often interacting issues (radiative forcing due to dust on snow events, warm spells, lack of winter snowfall, cloudiness); addressing those would substantially extend beyond the scope of this paper. To clarify our focus on the early season, we added the following text to the introduction:**
*"We focus on the initial timing of sufficient snow depth since the greatest demands for OSV recreation and potential ecological impacts occur between early and middle winter."*

4. In your abstract I suggest you talk about delayed onset date of SWEmin rather than the date increasing... the later is not intuitive to many people.

**Thank you for the suggestion, we have revised the text accordingly to highlight the delayed onset date. Because this is a key message of the paper, we are happy to continue to revise the text to ensure the correct message is clearly conveyed to readers if need be.**
*"Since 1985, median SWE$_{min}$ **onset has shifted later** by approximately two weeks. Potential proximal causes of **delayed onset** are investigated;…"*

Best regards, Ross Brown (ed)

---

## Author Comment (AC3) · 29 Nov 2018

**Author responses to: "Brief Communication: Early season snowpack loss and implications for over-snow vehicle recreation travel planning"**

Responses to reviewers are in **bold,** *new text is in italics* **(bold italics for emphasis)**
* * *
**Responses to Reviewer 2**
* * *
I appreciate the opportunity to review this brief communication, as there is very limited work on the implications of changing cryosphere for the multi-billion dollar over-snow vehicle (OSV) ('snowmobile') industry. This industry is far more at risk to cryosphere changes than the ski industry, which has much higher adaptive capacity through cost-effective snowmaking. I concur with G. Patterson that the paper is very well written and the comments on methods and interpretation, and will build on those remarks.

**Dear Dr. Scott,**
**We appreciate your constructive comments and positive feedback. Please find below responses and our revisions to address all of your comments. We have included your insightful comment about the greater risk to climate change the snowmobile community faces due to the absence of snowmaking in the introduction (please see the response to your first comment below).**

The dual data sources are very important to enable the exportability of the method to other regional markets across North America (mostly via reanalysis, as Snotel stations are limit in Eastern markets) that would allow inter-regional market comparisons.
A limitation to the paper is that literature review is not comprehensive and given how limited this literature is, it should be complete in my opinion. While I acknowledge space is limited in a brief communication, this will provide future authors with a complete and current state of knowledge to build on. This will also strengthen some of the threshold assumptions made in this paper and comparisons with other regional markets. Specifically, the authors should consider the work on snowmobile tourism under climate change that was completed in New England (Scott et al. 2008. Mitigation and Adaptation Strategies to Global Change, 13, 5-6, 577-59) and parts of Canada (McBoyle et al. 2007. Managing Leisure, 12, 4, 237-250) about 10 years ago.

**We agree and have included these citations (thank you for providing these) and relevant discussion in the introduction and discussion section. Specifically, we added two sentences summarizing the results of these two studies to the introduction:**
*"Due to the dependence on natural snowfall and reduced adaptive capacity compared to the ski community, which can use cost-effective snowmaking to augment the natural snowpack, over-snow vehicle (OSV) recreation is highly vulnerable to climate variability and change (Scott et al. 2007; Mcboyle 2007). Climate change projections for Canada and the northeastern United States under an aggressive greenhouse gas emissions scenario suggest that by the mid-21$^{st}$*

*century, OSV season lengths will be reduced by 50-100% in most areas (Mcboyle et al., 2007; Scott et al. 2007)."*

Discussion of impacts for visitor experience or economic impacts could be strengthened. Analyses of the impacts of recent record warm winters on the ski industry have revealed that shorter, more varied seasons result in increased congestion, which has adverse impacts on visitor experience (and thus economic surplus). The same impact is likely with OSV (particularly at trailheads) if demand remains stable.

**Thank you for the suggestion to add the concept of increased congestion and the resultant impact on high quality experiences to the discussion. We utilized recent results from Perry et al. (2018) to highlight these adverse impacts:**
*"A survey of the OSV community in Vermont found that reductions in the length of the winter season with sufficient snow coverage for OSV use were observed by 45% of respondents, with 74% of respondents decreasing their OSV use in response to low snow conditions (Perry et al., 2018). This survey also found that encounters with other recreationalists, including OSV users, detracted from a high-quality recreation experience."*

Have recent record warm winters revealed any impacts on visitor use patterns or increased impacts on landscapes/ecology?

**We are not aware of local (regionally-relevant) changes in visitor use (grouping this with economic impacts in an implicit sense) or ecological impacts, but we have added the latter to the concluding remarks sentence noting the need for additional studies on these topics (see bold italics below):**
*"Additional studies on achieving regionally-relevant minimum snow depths and better quantification of economic **and ecological** impacts from reduced snow cover area and duration will guide more robust travel management plans in national forests."*

**The work of Hagenstad et al. (2018) does provide insight regarding visitor use pattern change as a function of recent climate variability, and we have added sentences on this to the introduction:**
*"Skier visits are positively correlated to snowfall (Hagenstad et al., 2018) and we assume that such a correlation is consistent across winter recreation activities."*

*"The net effects of reduced season length, more congestion, and lower quality experiences result in lower economic benefits from consumer surplus, or the amount a person is willing to pay over the amount actually spent. For OSVs, consumer surplus is estimated to be approximately 61 USD day$^{-1}$ (Hagenstad et al., 2018)."*

The adaptation table is very useful for resource managers to consider appropriate responses. I fully agree with the authors that there is no one-size-fits-all approach, and that climate adaptation has to be informed by local circumstances and stakeholders. Other options the might be included in this table could include: (1) improve smoothness/durability of trailhead and corridor trails, so to require less snow and reduce impacts; (2) restrict access to marked trail areas in early/late season or during mid-season low snow periods; (3) if trail groomers are not used in this region, introduce them to improve the durability of corridor trails.

**Great suggestions, thank you. We have added suggestions (1) and (3) to Table 1 and have adjusted an existing adaptation measure to include the marked trails noted in suggestion (2). Our new table is as follows (and includes the suggestion from Reviewer 1 as well):**

| Adaptation Measure | Benefit(s) | Challenge(s) |
|---|---|---|
| *Requirement of minimum snow depth off trail, but not on roads/marked trails, or a lower minimum snow depth on roads/marked trails* | Allow OSV use even under extremely low snow conditions, limits resource damage in wildlands; grooming could be utilized to maximize snow depth on road | Preventing users from going off trail under low snow conditions; enforcement, resources required to obtain snow condition information |
| *Ensure high elevation access via a right-of-way* | During warmer/drier years, snow conditions are likely to be better (deeper snowpack) at higher elevation | User group conflicts; presence of Wilderness at high elevation; impacts to snow-dependent wildlife species; demand; parking |
| *Removal of blanket opening dates* | Prevents opening before $SWE_{min}$ achieved and will limit damage to landscape | Resources required to obtain snow condition information |
| *Identify corridors that collect/retain more snow* | During otherwise poor snow conditions, these areas may allow OSV recreation to occur, particularly at lower elevation areas | Need for data on these corridors |
| *Improve durability of trailhead and corridor trails* | Allows OSV recreation to occur when minimal snow exists thereby reducing negative impacts in high-use areas | Need for specific quantification of how to improve durability; potential permitting problems |
| *Trade-off: closure of low elevation/sensitive habitat for improved high elevation access* | Eliminate chance of damaging landscapes in low elevation regions, increase in the number of days/year that OSV recreation can occur by enhanced high elevation access | Need for collaboration between stakeholders/user groups to identify areas where compromise could occur; may be opposed by those who must travel much further for OSV use. |
| *Fee increases to enhance access and offset impacts from higher demand (i.e., restoration projects)* | Would provide for additional resources to monitor trailhead conditions, improve parking/bathrooms at trailheads, fund restoration projects and creation of low-snow OSV trails | Fees are generally opposed by members of the public. |
| *Additional grooming* | Allows additional area for OSV use when conditions are insufficient for off-trail use | Costs for grooming equipment and personnel, many OSV users are primarily interested in off-trail use |
| *Clear designation of non-motorized areas (i.e., signage)* | Reduces user conflicts by improving knowledge and awareness of areas open (or closed) to OSV use | Costs related to enforcement as well as installation and upkeep of signage |

**Table 1: Adaptation strategies to address loss of early winter snowpack for OSV recreation.**

**Based on reviewer 2's comments, we have added the following references:**

Hagenstad, M., Burakowski, E.A., and Hill, R. Economic contributions of winter sports in a changing climate, available at:
https://scholars.unh.edu/cgi/viewcontent.cgi?article=1190&context=ersc, (last accessed 25 November 2018), 2018.

Mcboyle, G., Scott, D., and Jones, B. Climate change and the future of snowmobiling in non-mountainous regions of Canada, Manag. Leisur., 12 (4), 237–250, https://doi.org/10.1080/13606710701546868, 2007.

Perry, E., Manning, R., Xiao, X., Valliere, W., and Reigner, N.: Social climate change: The advancing extirpation of snowmobilers in Vermont, J. Park Rec. Admin., 36, 31-51, https://dx.doi.org/10.18666/JPRA-2018-V36-I2-8307, 2018.

Scott, D., Dawson, J., and Jones, B. Climate change vulnerability of the U.S. Northeast winter recreation–tourism sector, Mitig. Adapt. Strat. Glob. Chang., 13 (5–6), 577–596, https://doi.org/10.1007/s11027-007-9136-z, 2008.